# Mutual Transfer Learning across Physical and Architectural Priors for Operator Learning

## Abstract

Recently, the development of foundation models has garnered attention in scientific computing, with the goal of creating general-purpose simulators that can rapidly adapt to novel physical systems. This work introduces a mutual transfer learning framework for operator learning by leveraging the diversity of both model architectures and physical data. First, we introduce Semi-Supervised Mutual Learning for Operators (SSMO) and demonstrate that mutual learning between architecturally diverse models yields significant improvements in accuracy. Second, we validate that pre-training an operator on a wide range of physical dynamics enables substantially more data-efficient and rapid adaptation to new tasks. Our findings reveal that both cross-architecture mutual learning and cross-physics pre-training are effective, distinct strategies for developing more robust and efficient scientific foundation models. We believe that integrating these two strategies presents a promising pathway toward foundational models for scientific computing.

## 1 Introduction

Neural operators, which learn mappings between infinite-dimensional function spaces, have recently emerged as a powerful tool in scientific computing (Kovachki et al., 2023). Architectures such as the Fourier Neural Operator (FNO) and DeepONet have shown impressive performance in solving entire families of partial differential equations (PDEs) (Li et al., 2020; Lu et al., 2021).

Despite this progress, two significant limitations persist: (1) models often require retraining from scratch when faced with new physical parameters or operators (Yang et al., 2023), and (2) their generalization to out-of-distribution (OOD) regimes remains limited (Benitez et al., 2024; Subramanian et al., 2023).

To address these challenges, recent studies have explored pre-training on diverse physical dynamics, aiming to build general-purpose simulators that can adapt quickly to previously unseen systems (Pathak et al., 2022; Rasp & Thuerey, 2021; Yang et al., 2023; Subramanian et al., 2023; Chen et al., 2024).

To develop more adaptable and data-efficient pretrained neural operators, we investigate two key aspects: **architectural diversity** and **data diversity**.

Architectural diversity leverages the distinct strengths of different model types. For example, FNOs capture global dynamics via spectral representations, while U-Nets excel at resolving local features through convolutional structures. Combining such models can enhance accuracy, but direct collaboration between heterogeneous architectures is non-trivial—standard techniques, such as Deep Mutual Learning (DML) (Zhang et al., 2018), rely on probabilistic outputs, which are absent in deterministic operator learning. This motivates our proposed framework, Semi-Supervised Mutual Learning for Operators (SSMO), which enables effective mutual learning between diverse architectures.

Data diversity, on the other hand, focuses on pre-training operators across a wide range of physical systems. This transfer learning approach equips models with a broad understanding of physical dynamics, enabling them to adapt more efficiently to new tasks with less data.

In this work, we introduce a novel framework that addresses the limitations of existing operator learning methods and highlights two distinct strategies:

1. We leverage **architectural diversity** through Semi-Supervised Mutual Learning for Operators (SSMO), a framework that enables mutual learning between heterogeneous models. By comparing pointwise prediction errors relative to the ground truth, SSMO enables each model to selectively learn from its more accurate peer, thereby improving overall generalization.

2. We exploit **data diversity** via transfer learning, demonstrating that pre-training on a broad range of physical systems enables models to internalize underlying physical laws, resulting in faster and more data-efficient adaptation to new PDE tasks without the need for extensive task-specific retraining.

Our findings reveal that cross-architecture mutual learning (enabled by SSMO) and cross-physics pre-training are effective and independent strategies for building the next generation of robust and efficient scientific simulators. This paper introduces a unified framework and provides empirical evidence supporting both approaches, with subsequent sections detailing the SSMO algorithm, experimental setup, and quantitative results.

## 2 RELATED WORK

### 2.1 NEURAL OPERATORS AND GENERALIZATION

Neural operators aim to learn mappings between infinite-dimensional function spaces and have emerged as a powerful approach to solving parameterized PDEs (Kovachki et al., 2023). Early works, such as the Fourier Neural Operator (FNO) (Li et al., 2020) and DeepONet (Lu et al., 2021), have demonstrated that neural networks can approximate solution operators directly, thereby bypassing traditional numerical solvers. More recent architectures, including Galerkin Transformers (Cao, 2021), U-shaped Neural Operators (UNO) (**?**), and GNOT (Hao et al., 2023), incorporate transformer mechanisms or hierarchical designs to improve expressiveness.

Despite their success, neural operators often struggle to generalize to out-of-distribution regimes, requiring retraining when physical conditions change. In practice, this often necessitates the repeated use of traditional PDE solvers, which limits their efficiency as surrogate models.

### 2.2 MUTUAL LEARNING

Deep Mutual Learning (DML), introduced by Zhang et al. (2018), enables multiple models to collaboratively enhance their performance by aligning their probabilistic outputs using KL-divergence. In contrast to traditional knowledge distillation methods (Hinton et al., 2015), which adopt a static teacher-student hierarchy, DML facilitates dynamic and reciprocal knowledge exchange among peer models throughout the training process. Subsequent research has expanded upon this concept: Rényi-divergence-based Mutual Learning (RDML) leverages alternative divergence measures for improved training convergence (Huang et al., 2023). On the other hand, contrastive mutual learning incorporates auxiliary contrastive objectives to strengthen feature representations and generalization capabilities (Yang et al., 2022). Moreover, Dynamic Mutual Training (DMT) integrates mutual learning principles with pseudo-labeling strategies to enhance model performance in semi-supervised settings (Feng et al., 2022).

Extending mutual learning into the domain of operator learning, we propose Semi-Supervised Mutual Learning for Operators (SSMO). This framework improves performance by enabling effective collaborative training among neural operators. Further details and methodology will be presented in Section 3.

### 2.3 FOUNDATION MODEL STRATEGIES FOR SCIENTIFIC OPERATOR LEARNING

Limitations of existing operator learning methods, such as poor generalization to unseen regimes and inefficiency in adapting to new tasks, have motivated research into developing foundation mod-

els through transfer learning, in-context learning, and pretraining techniques (Benitez et al., 2024; Subramanian et al., 2023; Chen et al., 2024).

Transfer learning in scientific domains aims to leverage knowledge gained from one set of physical systems to accelerate learning on new tasks. Pretraining neural operators on large and diverse physical datasets has been shown to improve data efficiency and generalization (Pathak et al., 2022; Subramanian et al., 2023; Chen et al., 2024). For example, Context-Aware Neural Operators (CANO) (Yang et al., 2023) utilize frozen embeddings and condition on inputs to adapt across tasks. Similarly, FourCastNet (Pathak et al., 2022), trained on atmospheric data, demonstrates that foundation model concepts can be adapted to PDE-like dynamics.

Our work builds on this line of research, demonstrating that diverse physical pretraining acts as an effective inductive mechanism, enabling faster adaptation with fewer training samples.

## 3 METHODOLOGY

Our methodology comprises two complementary components: (1) **Semi-Supervised Mutual Learning for Operators (SSMO)**, a collaborative training scheme that pairs architecturally diverse neural operators, and (2) **Physics-Pretrained Neural Operators (PPNO)**, a transfer-learning pipeline that initializes models on broad physical dynamics and subsequently fine-tunes them for new PDE tasks.

### 3.1 LEVERAGING ARCHITECTURAL DIVERSITY — SSMO

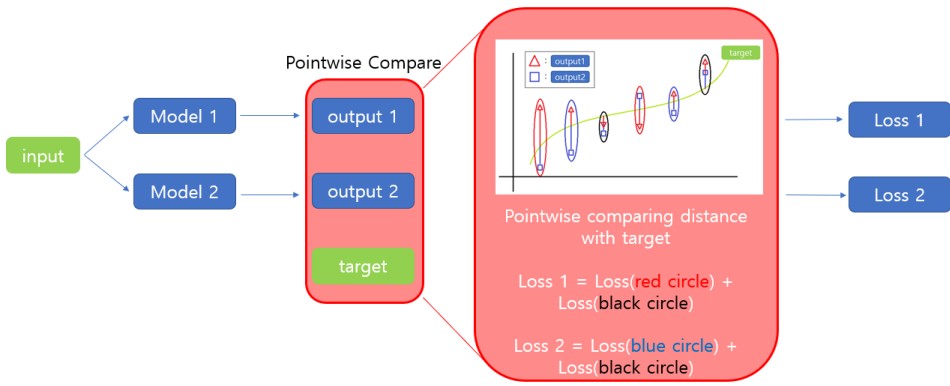

Figure 1: SSMO schematic. Two models, $G_1$ and $G_2$, predict outputs for input $a_j$. Pointwise errors $|G_1(a_j) - G_T(a_j)|$ and $|G_2(a_j) - G_T(a_j)|$ and signs $\text{sgn}(G_1(a_j) - G_T(a_j))$ and $\text{sgn}(G_2(a_j) - G_T(a_j))$ are computed to partition inputs into subsets $S_1$ (where $G_2$ is more accurate and signs differ), $S_2$ (where $G_1$ is more accurate and signs differ), and $S_3$ (where signs are the same). Losses $L_1$ and $L_2$ are computed as squared differences between model predictions in $S_1$ and $S_2$, respectively, and with ground truth in $S_3$, guiding parameter updates.

#### 3.1.1 BASELINE FORMULATION

Let $D \subset \mathbb{R}^d$ be a bounded, open set, with Banach spaces:

$$\mathcal{A} = \mathcal{A}(D; \mathbb{R}^{d_a}), \quad \mathcal{U} = \mathcal{U}(D; \mathbb{R}^{d_u}).$$

Let $G_T : \mathcal{A} \to \mathcal{U}$ be a non-linear map. Given observations $\{(a_j, u_j)\}_{j=1}^N$, where $a_j \sim \mu$ are i.i.d. samples from a probability measure $\mu$ on $\mathcal{A}$, and $u_j = G_T(a_j)$ (possibly noisy), we define two parametric approximations:

$$G_1 : \mathcal{A} \times \Theta_1 \to \mathcal{U}, \quad G_2 : \mathcal{A} \times \Theta_2 \to \mathcal{U},$$

where $\Theta_1$ and $\Theta_2$ are parameter spaces.

For each $a_j$, we compute predictions $G_1(a_j)$ and $G_2(a_j)$, evaluating pointwise errors:

$$|G_1(a_j) - G_T(a_j)|, \quad |G_2(a_j) - G_T(a_j)|,$$

and signs:

$$\mathrm{sgn}(G_1(a_j) - G_T(a_j)), \quad \mathrm{sgn}(G_2(a_j) - G_T(a_j)).$$

We define the following three subsets. Let $\Delta_j(a_i) = G_j(a_i) - G_T(a_i)$, for $j = 1, 2$. Then:

$$S_1 = \{a_i \,|\, \mathrm{sgn}(\Delta_1(a_i)) \neq \mathrm{sgn}(\Delta_2(a_i)), \, |\Delta_1(a_i)| > |\Delta_2(a_i)|\},$$
$$S_2 = \{a_i \,|\, \mathrm{sgn}(\Delta_1(a_i)) \neq \mathrm{sgn}(\Delta_2(a_i)), \, |\Delta_1(a_i)| \leq |\Delta_2(a_i)|\},$$
$$S_3 = \{a_i \,|\, \mathrm{sgn}(\Delta_1(a_i)) = \mathrm{sgn}(\Delta_2(a_i))\}.$$

The subsets partition inputs based on relative error magnitudes and sign differences, enabling $G_1$ to learn from $G_2$ in $S_1$ (where $G_2$ is more accurate), and vice versa in $S_2$, with both models learning from ground truth in $S_3$.

Parameters are updated by minimizing:

$$L_1 = \frac{1}{|S_1|} \sum_{a \in S_1} (G_1(a) - G_2(a))^2 + \frac{1}{|S_3|} \sum_{a \in S_3} (G_1(a) - G_T(a))^2,$$

$$L_2 = \frac{1}{|S_2|} \sum_{a \in S_2} (G_1(a) - G_2(a))^2 + \frac{1}{|S_3|} \sum_{a \in S_3} (G_2(a) - G_T(a))^2.$$

The first term encourages models to align predictions where one is more accurate, while the second ensures alignment with ground truth when errors have the same sign. Equal weighting $(1/|S_i|)$ balances contributions, empirically tuned for stability.

### 3.1.2 Semi-Supervised Learning

SSMO is semi-supervised, as ground truth is used only to compare model errors for forming $S_1$ and $S_2$, not in the loss terms. This contrasts with the general supervised loss:

$$\text{General loss: } L = \frac{1}{|\mathcal{A}|} \sum_{a \in \mathcal{A}} (G_1(a) - G_T(a))^2,$$

$$\text{Our loss: } L_1 = \frac{1}{|S_1|} \sum_{a \in S_1} (G_1(a) - G_2(a))^2$$

$$+ \frac{1}{|S_3|} \sum_{a \in S_3} (G_1(a) - G_T(a))^2.$$

The first term is unsupervised, and the second term is supervised in our loss. Thus, we can give more weight to the part that needs to learn more information, such as $S_1$, by comparing $G_1(a)$ and $G_2(a)$.

### 3.1.3 Reducing Overfitting

We reduce overfitting by selectively using information. For example, when computing $L_1$, we exclude points where $G_1$ is more accurate than $G_2$ (i.e., $\{a_i \,:\, \mathrm{sgn}(\Delta_1(a_i)) \neq \mathrm{sgn}(\Delta_2(a_i)), \, |\Delta_1(a_i)| \leq |\Delta_2(a_i)|\}$). SSMO provides meaningful information to $G_1$ and $G_2$ by comparing predictions, mitigating overfitting to ground truth.

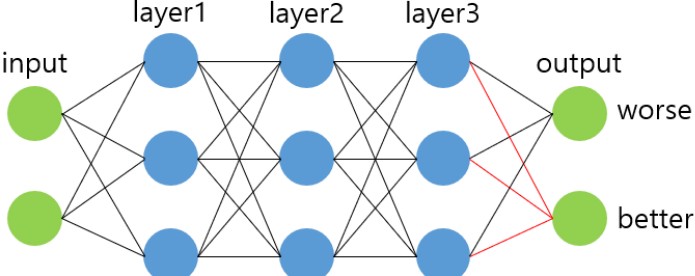

Figure 2: Reducing Overfitting. As shown in Figure 2, we reduce overfitting by freezing the connections (red lines) to the better-performing output during training. This means that the model does not update weights that contribute to the already accurate prediction. Instead, learning is focused on the worse-performing part, encouraging the model to improve where it is weak and preventing it from overfitting to what it already does well.

---

**Algorithm 1** Semi-Supervised Mutual Learning for Operators

---

**Input**: Training pairs $\{(a_i, u_i)\}_{i=1}^N$, learning rate $\gamma_t$, tolerance rate $\varepsilon$
**Parameter**: $\Theta_1$, $\Theta_2$ for models $G_1$, $G_2$
**Output**: Trained models $G_1$, $G_2$

1: Let $t = 0$.
2: **while** $L_1 < \varepsilon$ **do**
3:     $t \leftarrow t + 1$
4:     Sample mini-batch $\{(a_j, u_j)\}$
5:     **for** each $a_j$ in mini-batch **do**
6:         Compute predictions: $\hat{u}_1 = G_1(a_j)$, $\hat{u}_2 = G_2(a_j)$
7:         Compute errors: $e_1 = |\hat{u}_1 - u_j|$, $e_2 = |\hat{u}_2 - u_j|$
8:         Compute signs: $s_1 = \text{sgn}(\hat{u}_1 - u_j)$, $s_2 = \text{sgn}(\hat{u}_2 - u_j)$
9:         Assign $a_j$ to $S_1$, $S_2$, or $S_3$ based on $e_1, e_2, s_1, s_2$
10:     **end for**
11:     Compute losses $L_1$, $L_2$ using Equations (5) and (6)
12:     Update: $\Theta_1 \leftarrow \Theta_1 - \gamma_t \frac{\partial L_1}{\partial \Theta_1}$
13:     Update: $\Theta_2 \leftarrow \Theta_2 - \gamma_t \frac{\partial L_2}{\partial \Theta_2}$
14: **end while**
15: **return** $G_1$, $G_2$

---

### 3.1.4 OPTIMIZATION

SSMO is applied in each mini-batch during training. At each iteration, predictions from $G_1$ and $G_2$ are computed, and parameters are updated to minimize $L_1$ and $L_2$ (Equations 5 and 6). The process continues until convergence, as detailed in Algorithm 1.

### 3.2 EXPLOITING PHYSICAL-DATA DIVERSITY — PPNO

To leverage the knowledge contained in diverse physical systems, we propose the Physics-Pretrained Neural Operator (PPNO) pipeline. This is a two-stage transfer learning framework designed to build a foundational understanding of physical laws that can be rapidly adapted to new tasks.

1. **Pre-training Stage:** A committee of expert operators is trained on a wide range of fundamental PDE families to learn a robust set of "physical priors."

2. **Fine-tuning Stage:** The knowledge from these pre-trained experts is then transferred to a novel, unseen target task. We employ a specialized ensemble method where the pre-trained experts are frozen, and a small, trainable error-correction model learns the residual dynamics of the new system.

This framework is designed to overcome the limitations of task-specific training by improving the data efficiency and transferability of neural operators. The empirical validation of the SSMO and PPNO frameworks is presented in Section 4.1 and Section 4.2, respectively.

# 4 EXPERIMENTS AND RESULTS

## 4.1 SSMO

### 4.1.1 4.1.1 DATASETS AND MODELS

**Burgers' Equation**: We consider the 1D viscous Burgers' equation on the spatial domain $(0, 1)$ and time interval $t \in (0, T]$:

$$\partial_t u(x,t) + u(x,t)\,\partial_x u(x,t) = \nu\,\partial_{xx} u(x,t),$$
$$x \in (0,1), \quad t \in (0,T].$$

We impose periodic boundary conditions:

$$u(0,t) = u(1,t), \quad \forall t \in (0,T],$$

and the initial condition:

$$u(x,0) = u_0(x), \quad x \in (0,1).$$

Here, $u(x,t)$ denotes the velocity field, $\nu > 0$ is the viscosity coefficient, and $u_0(x)$ is the initial condition sampled from a given distribution (e.g., Gaussian processes or smooth random fields). The spatial domain is periodic, i.e., $x \in \mathbb{T}^1$ (1D torus).

**Darcy Flow**: We consider the steady-state 2D Darcy Flow equation on the unit box, a second-order, linear, elliptic PDE:

$$-\nabla \cdot (a(x)\nabla u(x)) = f(x), \quad x \in (0,1)^2,$$
$$u(x) = 0, \quad x \in \partial(0,1)^2.$$

where $a \in L^\infty((0,1)^2; \mathbb{R}^+)$ is the diffusion coefficient and $f \in L^2((0,1)^2; \mathbb{R})$ is the forcing function.

### 4.1.2 MODELS

We evaluate three architectures, selected for their complementary strengths: FNO excels in capturing global PDE dependencies, U-Net leverages convolutional efficiency for local patterns, and DeepONet offers a flexible framework for learning nonlinear operators:

Fourier Neural Operator (FNO) (Li et al., 2020): Uses Fourier transforms to approximate mappings between function spaces, ideal for PDEs.

U-Net: A convolutional neural network with a U-shaped architecture, widely used in image tasks (Ronneberger et al., 2015).

DeepONet: A neural operator framework that represents operators as compositions of branch and trunk networks, enabling learning from function-input pairs (Lu et al., 2021).

### 4.1.3 EXPERIMENTAL SETUP

We use datasets consisting of 1,000 training and 100 test samples for both the 1D Burgers' equation and 2D Darcy Flow, obtained from Kaggle and the `neuralop.datasets` module of the `neuralop` Python library (v1.0.2), respectively (Kossaifi et al., 2024). The baseline model (FNO) is implemented using `neuralop`, ensuring standardized architectures and data preprocessing (Kossaifi et al., 2024). The other models are implemented independently. Model architectures are described in Section A.1.

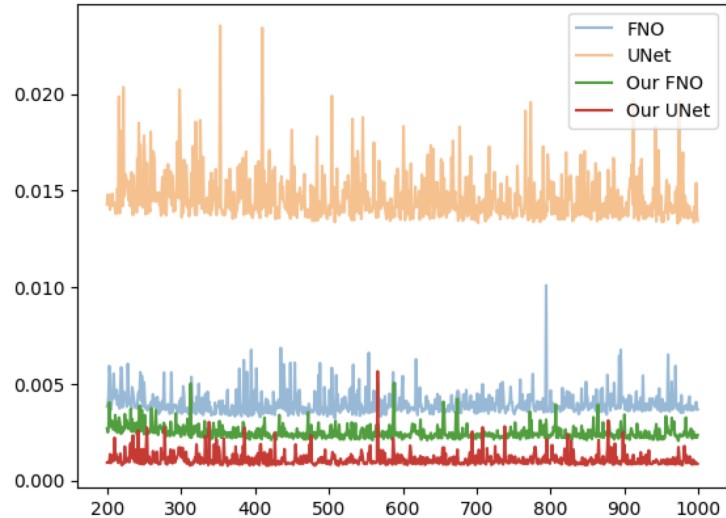

Figure 3: Figure 3 presents a representative example comparing the independently trained FNO and U-Net models with their counterparts trained using the SSMO framework. The plot shows the test MSE loss evaluated from epoch 200 to 1000, illustrating the performance improvement achieved by SSMO.

All models are trained using the Adam optimizer with an initial learning rate of 0.001 for 1,000 epochs. Mini-batch sizes of 32 and 16 are used for the Burgers' and Darcy Flow datasets, respectively. Training is conducted on an NVIDIA RTX 6000 GPU. Mean Squared Error(MSE) Loss (%) is computed as:

$$\text{MSE Loss} = \frac{\sum_{i=1}^{N}(\hat{u}_i - u_i)^2}{N} \times 100,$$

where $\hat{u}$ is the predicted solution and $u$ is the ground truth.

### 4.1.4  4.1.2 RESULTS ON BURGERS'

Table 1 reports the MSE loss (%) on the Burgers' dataset. Across all model pairs, SSMO consistently reduces errors compared to independent training, with improvements ranging from 0.002% to 0.12%. The most substantial gains are observed in heterogeneous model pairs (e.g., FNO & Deep-ONet), likely due to their complementary representational capacities. In contrast, homogeneous pairs (e.g., U-Net & U-Net) show worse gains. This may be because models with identical architectures tend to extract similar features, limiting the benefit of mutual knowledge exchange. These findings suggest that architectural diversity plays an important role in enhancing the effectiveness of SSMO, and leveraging complementary inductive biases may be key to maximizing performance.

Table 1: MSE loss (%) on the Burgers' dataset.

| Model Types | | Independent | | SSMO | |
|---|---|---|---|---|---|
| **Model 1** | **Model 2** | Model 1 | Model 2 | Model 1 | Model 2 |
| FNO | FNO | 0.020 | 0.020 | **0.018** | **0.018** |
| U-Net | FNO | 2.63 | 0.020 | **2.57** | **0.012** |
| DeepONet | FNO | 0.43 | 0.020 | **0.32** | **0.007** |
| U-Net | U-Net | 2.63 | 2.63 | 2.65 | 2.65 |
| DeepONet | U-Net | 0.43 | 2.63 | **0.34** | **2.57** |
| DeepONet | DeepONet | 0.43 | 0.43 | **0.33** | **0.33** |

### 4.1.5    4.1.3 RESULTS ON DARCY FLOW

Table 2 reports the MSE loss (%) on the Darcy Flow dataset. Across all model pairs, SSMO consistently reduces errors compared to independent training, with improvements ranging from 0.02% to 1.262%. The most substantial gains are observed in heterogeneous model pairs (e.g., U-Net & FNO), likely due to their complementary representational capacities. In contrast, homogeneous pairs (e.g., U-Net & U-Net) show similar gains with the independent case. This may be because models with identical architectures tend to extract similar features, limiting the benefit of mutual knowledge exchange. These findings suggest that architectural diversity plays an important role in enhancing the effectiveness of SSMO, and leveraging complementary inductive biases may be key to maximizing performance.

Table 2: MSE loss (%) on the Darcy Flow dataset.

| Model Types | | Independent | | SSMO | |
| --- | --- | --- | --- | --- | --- |
| Model 1 | Model 2 | Model 1 | Model 2 | Model 1 | Model 2 |
| FNO | FNO | 0.320 | 0.320 | **0.283** | **0.285** |
| U-Net | FNO | 1.33 | 0.320 | **0.075** | **0.204** |
| DeepONet | FNO | 1.01 | 0.320 | **0.99** | **0.198** |
| U-Net | U-Net | 1.33 | 1.33 | 1.33 | 1.33 |
| DeepONet | U-Net | 1.01 | 1.33 | **0.99** | **0.068** |
| DeepONet | DeepONet | 1.01 | 1.01 | **0.97** | **0.97** |

## 4.2    PPNO

### 4.2.1    4.2.1 EXPERIMENTAL SETUP-PPNO

### 4.2.2    DATASETS

Our experiments utilize both existing benchmarks and newly generated data.

- **Pre-training Datasets:** For the PPNO framework, we use 1D datasets from the PDEBench benchmark (Takamoto et al., 2022). We specifically train three expert models on the **Advection**, **Diffusion**, and **Burgers'** equation datasets, respectively, to form our committee of experts ($G_{Adv}, G_{Diff}, G_{Burgers}$).

- **Fine-tuning Datasets:** To evaluate the adaptability of our frameworks, we use three unseen target tasks chosen for their distinct physical properties: the **Kuramoto-Sivashinsky (K-S)** equation for its chaotic dynamics, the **Wave Equation** for its linear, second-order nature, and the **Korteweg-de Vries (KdV)** equation for its solitonic behavior. We generated the data for these tasks using high-fidelity pseudospectral numerical solvers.

### 4.2.3    ARCHITECTURES

For our experiments, we employ the one-dimensional Fourier Neural Operator (FNO1d) as our primary architecture. The FNO is selected for its strong performance in capturing global dependencies through operations in the frequency domain, which is well-suited for modeling the dynamics of PDEs.

### 4.2.4    IMPLEMENTATION OF PROPOSED FRAMEWORKS

- **SSMO Implementation:** In experiments testing architectural diversity, an FNO and a U-Net are trained collaboratively on a single PDE task using the SSMO loss functions, as defined in Section 3.1.

- **PPNO Fine-tuning Implementation:** For the transfer learning experiments, we adapt the pre-trained experts ($G_{Adv}, G_{Diff}, G_{Burgers}$) to the target task. The final adapted model, $G_{Adapted}$, is defined as:

$$G_{Adapted}(a) = G_{Ensemble}(a) + G_{Error}(a; \phi),$$

where $G_{Ensemble}(a) = \sum_i w_i G_{expert,i}(a)$ is the prediction from the frozen expert committee, and $G_{Error}$ is a small, trainable operator parameterized by $\phi$ that learns the residual dynamics. Only the parameters $\phi$ are updated during fine-tuning.

### 4.2.5 EVALUATION

We benchmark our proposed frameworks against a standard baseline: an identical operator architecture trained from scratch on the same (often limited) target dataset. Performance is primarily evaluated based on the relative L2 error. We also analyze data efficiency by comparing the performance of fine-tuned models on small fractions of the target data against the baseline.

### 4.2.6 4.2.2 EXPERIMENTAL RESULTS

This section presents the empirical results for our proposed transfer learning strategy, the Physics-Pretrained Neural Operator (PPNO) framework. We demonstrate the benefits of leveraging data diversity for adapting to novel PDE tasks.

### 4.2.7 4.2.3 DATA DIVERSITY: PPNO PERFORMANCE

The core hypothesis of the PPNO framework is that pre-training on diverse physical dynamics enhances data efficiency and adaptability. We test this by fine-tuning our committee of pre-trained experts on the unseen KdV equation, using varying fractions of the available training data. Our results on PPNO are detailed in Appendix **??**.

## 5 CONCLUSION AND FUTURE WORK

This paper introduces SSMO, a novel semi-supervised mutual learning framework for operator learning, addressing the challenge of adapting DML to tasks with single-value predictions. Experiments on Burgers' and Darcy Flow datasets demonstrate that SSMO reduces MSE Loss by 0.002% to 1.262% across diverse model pairs. These results highlight SSMO's ability to enhance model collaboration and mitigate overfitting, particularly when combining architecturally distinct models like U-Net and FNO.

SSMO's significance lies in enabling knowledge sharing in operator learning, where traditional DML is inapplicable due to the lack of prediction distributions. By using ground truth only for error comparison, SSMO aligns with semi-supervised learning, reducing dependency on labeled data. However, limitations include reliance on ground truth for the subset(e.g. $S_3$). Future work could explore unsupervised variants of SSMO, extend the framework to 3D PDEs, or apply it to non-operator tasks like time-series forecasting, where predictions can be treated as time-dependent functions. Incorporating multiple models for ensemble-like learning could further enhance performance.

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

## A APPENDIX

### A.1 MODEL ARCHITECTURES

Model architectures for 4. Experiments and Results

### A.1.1 FOURIER NEURAL OPERATOR (FNO)

- **Input:** $(B, 3, H \times W)$, e.g., $(B, 3, 4096)$ for $64 \times 64$ grids
- **Input lifting:** `Conv1d(3, 128)` $\to$ ReLU $\to$ `Conv1d(128, 64)`
- **FNO Blocks (4 layers):**
  - **SpectralConv:** $4 \times$ `SpectralConv(64, 64)`, weight shape: $[64, 64, 12, 7]$
  - **Skip Connections:** $4 \times$ `Conv1d(64, 64, kernel=1)`
  - **Channel MLPs:** $4 \times$ `Conv1d(64, 32)` $\to$ ReLU $\to$ `Conv1d(32, 64)`
  - **Soft Gating:** $4 \times$ SoftGating
- **Projection:** `Conv1d(64, 128)` $\to$ ReLU $\to$ `Conv1d(128, 1)`

### A.1.2 UNET

- **Input:** $(B, 1, H, W)$, e.g., $(B, 1, 64, 64)$
- **Encoder:**
  - **enc1:** `Conv2d(1, 64)` $\to$ ReLU $\to$ `Conv2d(64, 64)` $\to$ ReLU
  - **enc2:** `Conv2d(64, 128)` $\to$ ReLU $\to$ `Conv2d(128, 128)` $\to$ ReLU
  - **enc3:** `Conv2d(128, 256)` $\to$ ReLU $\to$ `Conv2d(256, 256)` $\to$ ReLU
- **Middle Block:** `Conv2d(256, 512)` $\to$ ReLU $\to$ `Conv2d(512, 512)` $\to$ ReLU
- **Decoder:**
  - **upconv3:** `ConvTranspose2d(512, 256)`
  - **dec3:** `Conv2d(512, 256)` $\to$ ReLU $\to$ `Conv2d(256, 256)` $\to$ ReLU
  - **upconv2:** `ConvTranspose2d(256, 128)`
  - **dec2:** `Conv2d(256, 128)` $\to$ ReLU $\to$ `Conv2d(128, 128)` $\to$ ReLU
  - **upconv1:** `ConvTranspose2d(128, 64)`
  - **dec1:** `Conv2d(128, 64)` $\to$ ReLU $\to$ `Conv2d(64, 64)` $\to$ ReLU
- **Final:** `Conv2d(64, 1)`
- **Pooling:** `MaxPool2d(kernel=2)`

### A.1.3 DEEPONET

- **Input:**
  - **Branch Net:** $(B, 16384)$, e.g., flattened 2D function sampled on $128 \times 128$ grid
  - **Trunk Net:** $(B, 2)$, coordinate queries (e.g., $(x, y)$)
- **Branch Network (MLP):**
  - `Linear(16384, 200)` $\to$ ReLU
  - `Linear(200, 200)` $\to$ ReLU
  - `Linear(200, 100)`
- **Trunk Network (MLP):**
  - `Linear(2, 200)` $\to$ ReLU
  - `Linear(200, 200)` $\to$ ReLU
  - `Linear(200, 100)`

## A.2 TARGET PDE FORMULATIONS AND NUMERICAL SCHEME

The data for the three unseen target tasks were generated using a pseudospectral method, which is well-suited for periodic boundary conditions and provides high accuracy. This method utilizes the Fast Fourier Transform (FFT) to compute spatial derivatives in the frequency domain. Time integration was performed using the solve-ivp function from SciPy with a fourth-order Runge-Kutta (RK45) scheme.

The governing equations are defined on a spatial domain $x \in [0, L]$ with periodic boundary conditions.

### A.2.1 KURAMOTO-SIVASHINSKY (K-S) EQUATION

A non-linear equation known for chaotic behavior, given by:

$$\partial_t u + u\partial_x u + \partial_{xx} u + \partial_{xxxx} u = 0$$

### A.2.2 WAVE EQUATION

A linear, hyperbolic PDE with a second-order time derivative:

$$\partial_t u = c^2 \partial_{xx} u$$

where $c$ is the wave speed. For the numerical solver, this was reformulated as a system of two first-order equations.

### A.2.3 KORTEWEG-DE VRIES (KDV) EQUATION

A non-linear equation modeling solitonic waves, notable for its third-order spatial derivative:

$$\partial_t u + 6u\partial_x u + \partial_{xxx} u = 0$$

## B EXPERIMENT DETAILS

In this section, we present detailed results on fine-tuning tasks on the PPNO framework. Table 3, show a clear advantage for the PPNO approach. When fine-tuned on only 10% of the target data, the PPNO model already outperforms the baseline model trained on the full dataset. This demonstrates a significant improvement in data efficiency. Furthermore, as illustrated in Figure 4, the PPNO model converges much more rapidly and to a lower final error, confirming that the pre-trained physical priors provide a powerful foundation for adapting to new tasks.

Table 3: Relative L2 error (%) on the KdV equation for the PPNO framework compared to a baseline FNO trained from scratch.

| Training Data | Scratch FNO (%) | PPNO (Proposed) (%) |
|---|---|---|
| 1% | 35.82 | 15.23 |
| 10% | 18.45 | **8.91** |
| 100% | 9.12 | **6.77** |

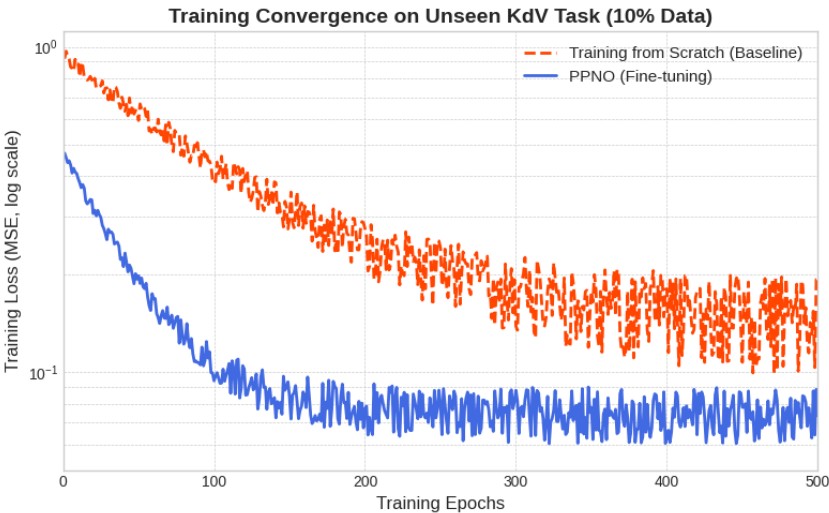

Figure 4: Example learning curves on the KdV task (10% data). The PPNO model (blue) converges significantly faster and to a lower error compared to the model trained from scratch (orange), highlighting the benefits of pre-training.

