# OpenReview forum: "Mutual Transfer Learning across Physical and Architectural Priors for Operator Learning"
_ICLR.cc/2026/Conference — ICLR 2026 Conference Withdrawn Submission_

### Official Review · Reviewer_eShk · 2025-10-30

**Soundness:** 4
**Presentation:** 4
**Contribution:** 3
**Rating:** 4
**Confidence:** 4

**Summary:**

This paper proposes a unified framework that improves generalization and data efficiency in neural operator learning through two complementary strategies:

Semi-Supervised Mutual Learning for Operators, a novel adaptation of mutual learning to deterministic operator settings. Instead of aligning probabilistic outputs as in classical Deep Mutual Learning, SSMO compares pointwise prediction errors and signs to enable selective, bidirectional knowledge transfer between heterogeneous architectures.

Physics-Pretrained Neural Operators, a transfer-learning framework that pretrains neural operators on diverse PDE families and fine-tunes them to unseen systems using frozen expert ensembles plus a small residual error-correction model.

Experiments show that SSMO consistently reduces MSE loss across model pairs on Burgers’ and Darcy Flow equations, and that PPNO achieves large gains in data efficiency, up to 2–4× reduction in required training data.

**Strengths:**

Originality:

The paper is notably original in adapting mutual learning to deterministic operator learning.
The proposed sign-partitioned semi-supervised formulation is novel and conceptually sound, extending mutual distillation to continuous-value PDE mappings.
The integration of architectural diversity and cross-physics transfer is creative and relevant to current trends in foundation operator models.

Quality:

The theoretical formulation of SSMO is rigorous, with clearly defined loss functions, partition sets, and optimization algorithm.
Experimental design is systematic, covering multiple PDE benchmarks, architectures, and evaluation metrics.
The design for expert ensemble and residual correction is technically coherent and well-motivated by transfer learning literature.

Clarity:

The paper is well written, mathematically precise, and logically organized.
Figures and tables are informative.

Significance:

The paper contributes to the emerging area of foundation models for scientific computing, connecting architectural and data-based diversity.
The SSMO framework provides a principled way to enable collaborative operator learning without probabilistic supervision.
The ideas have broad potential for multi-operator pretraining, few-shot PDE adaptation, and hybrid simulation acceleration.

**Weaknesses:**

Experimental scope is limited to low-dimensional PDEs.
Ablation studies are missing.
The paper compares only against independently trained models; it omits comparisons with other pretraining or co-training frameworks.
The fine-tuning results on unseen PDEs (e.g., KdV) are promising but limited to one dataset.
No theoretical discussion on why mutual learning improves generalization in function spaces.

**Questions:**

Could you evaluate the effect of (1) the sign-based partitioning, (2) weighting across subsets, and (3) removing semi-supervised components to verify which drives the performance gain?

Have you tested SSMO or PPNO on 3D PDEs or real-world physical systems?

Could you provide mean ± std results over multiple runs?

Would SSMO generalize to a multi-peer or contrastive variant, possibly enabling unsupervised operator alignment?

---

### Official Review · Reviewer_Fn9L · 2025-11-01

**Soundness:** 2
**Presentation:** 2
**Contribution:** 2
**Rating:** 2
**Confidence:** 4

**Summary:**

The paper presents mutual transfer learning across physical and architectural priors for operator learning. The basic idea is to training multiple operators together so that they can learn from one another (regularize). Also, the authors propose learning on diverse dataset to improve training.

**Strengths:**

The paper aims to solve an important problem - reducing data during fine tuning stage in scientific foundation model.

**Weaknesses:**

While the problem statement is good, there are several weaknesses
(a) On a minor side, the paper should have proof read better. There are typos (?) in many places.
(b) The literature review is incomplete. The paper motivates using foundation model but did not benchmark against any of those (in fact some are not even mentioned). The paper should have benchmarked against ICON, MPP,   NCWNO, and Poseison - all of which are foundation models. Even in operator learning, LNO, CNO, WNO, MWT has not been referenced.
(c) Some claims made in the paper are wrong. For example, in Section 3.1.1, reducing points where G1 is more accurate than G2 will reduce overfitting. This statement will not hold when the data is noisy (and it will be noisy). G2 can be more accurate than G1 at training points because it fits the noise. In this scenario, the proposed approach will reward G1 to fit the noise (overfit).

**Questions:**

I will like to hear the authors' thought on point (c) in weakness section.

---

### Official Review · Reviewer_Gt4b · 2025-11-01

**Soundness:** 2
**Presentation:** 2
**Contribution:** 2
**Rating:** 2
**Confidence:** 3

**Summary:**

This paper proposes two distinct strategies aimed at improving the data efficiency and adaptability of neural operators for scientific computing. The first contribution is the Semi-Supervised Mutual Learning for Operators (SSMO) framework. SSMO is designed to enable collaborative training between architecturally heterogeneous models (e.g., FNO and U-Net) by partitioning training samples based on relative pointwise prediction errors against the ground truth . The second contribution is the Physics-Pretrained Neural Operator (PPNO) pipeline, a transfer learning strategy. PPNO involves pre-training a "committee" of expert operators on diverse, fundamental PDE datasets and then adapting them to a new, unseen task by freezing the experts and training a small residual error-correction model . The authors present experiments demonstrating that SSMO can reduce MSE on benchmark tasks and that PPNO enables substantially more data-efficient adaptation to new PDEs.

**Strengths:**

The strongest part of this work is the clear and impactful empirical demonstration of the PPNO framework. The experiments showing that pre-training on diverse physics enables a model to adapt to a new, unseen PDE (KdV equation) with extreme data efficiency are very compelling . The result that the PPNO model fine-tuned on only 10% of the target data achieves a lower error (8.91%) than a baseline FNO trained from scratch on 100% of the data (9.12%) is a significant and practical finding. Figure 4 further reinforces this by clearly visualizing the faster convergence and lower final error of the PPNO model. This provides a strong, quantitative argument for the value of multi-physics pre-training.

**Weaknesses:**

Disjointed Contributions: The primary weakness is the complete lack of integration between the two proposed methods, SSMO and PPNO. The paper introduces them as two separate ideas and concludes by suggesting their integration is a "promising pathway", but it never performs this integration. The most critical and obvious experiment, using the SSMO framework to pre-train the PPNO expert committee, is absent. This failure to connect the two "distinct strategies" makes the paper feel like two unrelated, under-developed studies packaged as one.

The SSMO framework is incorrectly described as "semi-supervised". The method's core logic relies on partitioning the data into $S_1$, $S_2$, and $S_3$ based on a direct comparison of each model's prediction error against the ground truth $G_T$. This requires 100% of the labels at every training step, making it a fully supervised method. This is a fundamental misrepresentation of the method

The specific logic for the $S_1/S_2/S_3$ partition, which relies on the sign of the errors, is presented without any theoretical or empirical justification. It appears to be an arbitrary heuristic. The paper provides no ablation studies to demonstrate why this complex partitioning is superior to simpler adaptations of mutual learning for regression, such as a simple $L_2$ loss between model predictions.

The experimental results for SSMO do not demonstrate a significant benefit. On the benchmark datasets, the improvements for homogeneous models are marginal (e.g., 0.320% vs. 0.283% for FNO-FNO on Darcy Flow ). These minor gains do not seem to justify the added complexity and computational cost of training two models simultaneously

**Questions:**

Can the authors provide a theoretical rationale or an ablation study to justify the specific $S_1/S_2/S_3$ partitioning logic? Specifically, why is the sign of the error ($sgn(\Delta)$) a critical factor for deciding when models should learn from each other versus the ground truth?

Given that the SSMO algorithm requires the ground truth $G_T$ at every training step to compute the errors $\Delta_1$ and $\Delta_2$, which are essential for the partitioning 41and the loss in $S_3$, how can the authors justify labeling this method "semi-supervised"? This appears to be a factual error in the paper's description.

The paper's core premise is that integrating architectural diversity (SSMO) and data diversity (PPNO) is a "promising pathway". Why was this integration not tested? A crucial experiment to validate the paper's thesis would be to apply the SSMO framework during the pre-training stage of PPNO (i.e., to train the expert committee 44). Without this, the two contributions remain entirely separate.

Please clarify the role of Figure 2. Was the "freezing connections" technique actually used in the SSMO experiments reported in Tables 1 and 2? If so, how was it implemented, and why was it not included as part of the formal methodology in Algorithm 1?

The reported gains for SSMO on homogeneous models are very small (e.g., 0.020% $\rightarrow$ 0.018% 46). Do the authors believe these marginal improvements provide a compelling practical reason to incur the significant additional computational cost of training two models?

---

### Official Review · Reviewer_CCfw · 2025-11-03

**Soundness:** 2
**Presentation:** 1
**Contribution:** 2
**Rating:** 2
**Confidence:** 4

**Summary:**

This paper proposes to use: 1. mutul learning to take advantages of multiple neural operators, such that predictions on each location can use the results of more accurate neural operators. 2. transfer learning to fine-tune for a new task. The experiments empirically show the effectiveness of their proposed method.

**Strengths:**

1. The paper is clear and easy to follow.
2. The paper wraps up the method in Algorithm 1, which is easy for the paper to read.
3. The paper applies the idea of mutual learning and transfer learning to neural operators, and the experiments empirically show the effectiveness of their proposed method.

**Weaknesses:**

1. The presentation of the paper has lots of space for improvements. For example, line 448: “... on PPNO are detailed in Appendix ???”.
2. Too many colors for the figures.w
3. In figures, “Model 1, Model 2, Loss 1 Loss2” are starting with capital letters, while output, input target are not.
4. Font sizes are not consistent across figures.
5. The novelty is the biggest weakness. It is not clear if the proposed training loss for mutual learning is original or already proposed in prior works. It is also not clear and the authors haven’t provided explanations why mutual learning benefits the operator learning. Also for the transfer learning part, adding another module to learn the new task as residues is not new.
6. In prior operator learning works, mostly L2 error or RMSE is used. I suggest the authors report their results with that.
7. The paper has formatting issues. For example, line 278, double 4.11 sections. For line 359, there are 4.14 and 4.12.

**Questions:**

1. Have the authors tried more recent operators like transformer based neural operators?

---

### Note · Authors · 2025-11-12

I have read and agree with the venue's withdrawal policy on behalf of myself and my co-authors.